# Phenolic Acids and Flavonoids Play Important Roles in Flower Bud Differentiation in *Mikania micrantha*: Transcriptomics and Metabolomics

**DOI:** 10.3390/ijms242316550

**Published:** 2023-11-21

**Authors:** Ling Pei, Yanzhu Gao, Lichen Feng, Zihan Zhang, Naiyong Liu, Bin Yang, Ning Zhao

**Affiliations:** 1College of Life Sciences, Southwest Forestry University, Kunming 650224, China; peiling202204@163.com (L.P.); gaoyanzhu98@163.com (Y.G.); flc225169@163.com (L.F.); dondieee@163.com (Z.Z.); 2Key Laboratory of Forest Disaster Warning and Control of Yunnan Province, Southwest Forestry University, Kunming 650224, China; naiyong.liu@swfu.edu.cn

**Keywords:** *Mikania micrantha*, flower bud differentiation, transcriptomic, metabolomic, phenolic acid, flavonoid

## Abstract

*Mikania micrantha* is a highly invasive vine, and its ability to sexually reproduce is a major obstacle to its eradication. The long-distance dissemination of *M. micrantha* depends on the distribution of seeds; therefore, inhibiting *M. micrantha* flowering and seed production is an effective control strategy. The number of blooms of *M. micrantha* differs at different altitudes (200, 900, and 1300 m). In this study, we used a combination of metabolomics and transcriptomics methods to study the patterns of metabolite accumulation in the flower buds of *M. micrantha*. Using LC-MS/MS, 658 metabolites were found in the flower buds of *M. micrantha* at three different altitudes (200, 900, and 1300 m). Flavonoids and phenolic acids were found to be the main differential metabolites, and their concentrations were lower at 900 m than at 200 m and 1300 m, with the concentrations of benzoic acid, ferulic acid, and caffeic acid being the lowest. The biosynthesis pathways for flavonoids and phenolic compounds were significantly enriched for differentially expressed genes (DEGs), according to the results of transcriptome analysis. The production of flavonoid and phenolic acids was strongly linked with the expressions of phenylalanine ammonia-lyase (*PAL*), caffeoyl-CoA O-methyltransferase (*COMT*), and 4-coumarate-CoA ligase (*4CL*), according to the results of the combined transcriptome and metabolome analysis. These genes’ roles in the regulation of distinct phenolic acids and flavonoids during *M. micrantha* bud differentiation are still unknown. This study adds to our understanding of how phenolic acids and flavonoids are regulated in *M. micrantha* flower buds at various altitudes and identifies regulatory networks that may be involved in this phenomenon, offering a new approach for the prevention and management of *M. micrantha*.

## 1. Introduction

*Mikania micrantha* Kunth (Asteraceae) is a perennial herbaceous or woody vine of the *Mikania* Willd in the Asteraceae family. It is one of the most destructive weeds in the world [1,2]. *M. micrantha* is more adept in covering, diffusing, and rejecting than many other weeds. Through nutrient competition and covering, it can kill grass and shrub layers in the invaded region, as well as climb and cover trees and kill hosts [3]. *M. micrantha*, as a noxious invasive plant, has seriously harmed crops by lowering their quality, which has resulted in decreased income for locals. This vine has impacted local economic development and the growth of agricultural and forestry crops [4,5]. For nearly 30 years, researchers have adopted various physical, chemical, and biological control methods, but these methods have not effectively stopped the spread of *M. micrantha* [6,7,8]. The fundamental reason for this failure is the large number of flowers produced by *M. micrantha*, the flower biomass of which accounts for 38% to 43% of its total biomass [9]. These flowers produce a large number of seeds with crown hairs, which can spread over long distances, hindering control efforts [10,11]. *M. micrantha* is capable of reproduction and can reproduce both asexually and sexually, with its seeds being the main means of long-distance dispersal [12]. In this stage, there is an urgent need to tap into a quick and effective control measure. The suppression of the sexual reproduction of *M. micrantha* would be an effective method of controlling its rapid reproduction. Thus, controlling *M. micrantha*’s flowering and seed production is crucial for its management and defense.

During the transition from the nutritive to the reproductive growth stage of the plant, the apical meristematic tissue changes from the nutritive to the reproductive stem end, forming the flowering organ. The formation of plant flowering organs is a sign of plant flowering, and the time of plant flowering is crucial for plant growth, development, and reproduction [13,14]. Phenolic metabolites can affect the floral bud differentiation of plants: flavonoids and flavonols are evident in the heads of *Chrysanthemum morifolium* when waterlogging stress occurs in different stages of flower bud differentiation [15]. According to Ishimaru et al., the accumulation of chlorogenic acid is closely related to stress-induced flowering [16]. Oota et al. proposed that the free hydroxyl group of benzoic acid facilitates the induction of flowering [17]. The content of polyphenols, including quercetin, catechin, anthocyanins, and phenolic acids, in rose changes during flower-open development [18,19].

Using transcriptome sequencing (RNA-Seq) technology, genes associated with metabolite production can be identified [20]. To further understand metabolite synthesis and accumulation patterns, researchers have used plant metabolomics to qualitatively and quantitatively analyze the molecular metabolites in plants [21]. In recent years, metabolomics, in combination with transcriptomics, has provided a powerful tool for the study and characterization of the molecular basis of mechanisms in many plants [22]. For example, Lang et al., through the transcriptome and metabolomic analysis of a Rubellis flower, described the metabolism of flavonoids and anthocyanins in this early angiosperm plant [23].

In Yunnan Province, Liang et al. found differences in the distribution of *M. micrantha* at altitudes of 200, 900, and 1300 m. The ecological factors at these three altitudes substantially differed, but no notable differences were found among the three in terms of annual precipitation, relative humidity, or storm runoff. However, among the environmental factors related to altitude, temperature may considerably impact the ecological adaptability of *M. micrantha*. Light may also strongly affect the bud differentiation of *M. micrantha* [24]. We found that the numbers of florets and inflorescences at 900 m were significantly higher than they were at 200 and 1300 m. In this study, we investigated the differences in the gene expression and metabolics of *M. micrantha* at different altitudes using transcriptomic and metabolomic analyses. We aimed to gain insights into the genetic associations among the small-molecule metabolites to understand the key metabolites affecting flower bud differentiation in *M. micrantha*. We examined the relationship between gene transcript levels and metabolite accumulation in an effort to identify metabolites that affect flower bud differentiation in order to discover new methods of stopping the spread of *M. micrantha*.

## 2. Results

### 2.1. Metabolome Analysis

In this study, principal component analysis (PCA) was employed to assess the overall differences in the metabolite profiles among the groups and the variability within each group. The PCA results revealed that the first two principal components, PC1 and PC2, explained 44.2% and 32.9% of the total variance observed in the samples, respectively. Together, PC1 and PC2 accounted for a cumulative contribution rate of 77.1% (Figure 1A). Based on the PCA results, the samples were found to form distinct clusters, suggesting noticeable metabolite differences among the *M. micrantha* at different altitudes (Figure 1A). The analysis of the widely targeted metabolome was performed on *M. micrantha* flower buds at three altitudes using an LC-ESI-MS/MS system. A total of 658 metabolites were obtained from all the samples, which we classified (Appendix A, Figure 1B). Among them, flavonoids were the most abundant, with 161, followed by lipids and phenolic acids, with 94 and 78, respectively. The results of the hierarchical clustering analysis of flavonoids and phenolic acids revealed that their occurrence substantially differed at different altitudes (Appendix A, Figure 1C,D). The results indicated that the metabolites in the flower buds of *M. micrantha* widely differed at the three different altitudes.

In this study, a total of 107 differentially expressed metabolites (DEMs) were identified in the comparison between E2 and E9, with 66 being upregulated and 41 being downregulated. In the comparison between E13 and E9, 53 DEMs were identified, with 22 upregulated and 31 downregulated. The identified DEMs were categorized into different groups, including phenolic acids (Figure 1B), flavonoids, and other metabolites. Among these categories, flavonoids and phenolic acids were the most abundant, suggesting that they play an important role in bud differentiation in M.micrantha. In addition, the results of our cluster analysis of differential flavonoid metabolites and differential phenolic acid metabolites (Figure 1C,D) revealed that the contents of phenolic acids and flavonoids, e.g., 5-O-Caffeoylshikimic acid and Limocitrin-3-O-glucoside, were lower in *M. micrantha* buds at 900 m than at 200 and 1300 m, which suggests that the accumulation of most flavonoids and phenolic acids in *M. micrantha* buds is reduced at 900 m, possibly affecting the bud differentiation of *M. micrantha*. To gain deeper insight into the metabolic pathways involved, we mapped the enriched metabolites in the two comparison groups to the KEGG pathways. Specifically, the first 20 pathways were analyzed for each group (Figure 2). The results displayed significant enrichment of the DEMs in several pathways, including metabolic pathways, the biosynthesis of secondary metabolites, flavonoid biosynthesis, and phenylpropanoid biosynthesis. These findings highlight the importance of these pathways in the regulation and biosynthesis of secondary metabolites, particularly flavonoids and phenolic acids. This study sheds light on the differential expression and regulation of metabolites, particularly those of flavonoids and phenolic acids, during the bud differentiation phase of *M. micrantha*.

### 2.2. Transcriptome Sequencing

The transcriptome sequencing results of the *M. micrantha* flower buds sampled from the three altitudes obtained using RNA-Seq are shown in Appendix A. We had three biological replicates for each developmental stage, all with Q20 values greater than 97.00% and average percentages of 93.96% and 43.43% for Q30 and GC, respectively. Therefore, the transcriptome sequencing produced high-quality data. The clean reads were annotated from the Nr, GO, KEGG, Pfam, eggNOG, COG, KOG, and Swissport databases. The gene expression profiles of the nine analyzed samples were analyzed using PCA, and the results showed good within-group reproducibility and high between-group variability for the samples, indicating the reliability of the transcriptome data, which ensured the reliability of the subsequent differential gene analysis.

### 2.3. Analysis of DEGs

To elucidate the function of the DEGs, two comparison groups of DEG enrichment in the 20 most significant KEGG metabolic pathways based on the FPKM values were analyzed. The DEGs were mostly concentrated in metabolic pathways, the biosynthesis of secondary metabolites, flavonoid biosynthesis, and phenylpropanoid biosynthesis pathways, according to the KEGG enrichment analysis of the total DEGs (Figure 3). Among the analyzed KEGG pathways, the enrichment of the flavonoid and phenylacetone biosynthetic pathways was found to be especially significant. This indicates that these pathways play an important role in the observed metabolic changes. The expression profiles of the DEGs involved in the flavonoid biosynthesis pathway revealed certain patterns upon further investigation. Compared with E9, DEGs were expressed at higher levels at E2 and E13. This suggests that the regulation of flavonoid biosynthesis in *M. micrantha* is more active at 200 and 1300 m and suppressed at 900 m. At E2 and E13, the expression levels of the DEGs were higher than those at E9. Similar trends were observed in the expression levels of the DEGs involved in the phenylpropanoid biosynthesis pathway. Compared with 900 m, the plants at E2 and E13 exhibited higher levels of gene expression. The expression levels of the DEGs were higher at E2 and E13 than at E9. This indicates that phenylpropanoid biosynthesis is also more active at altitudes of 200 and 1300 m. The upregulation of DEGs in these pathways at 200 and 1300 m indicates the importance of their regulated metabolites in the bud differentiation process.

### 2.4. Conjoint Analysis of Transcriptome and Metabolome

Based on the results of the differential metabolite and differential gene enrichment analyses, the enrichment level of the pathways with both differential metabolites and differential genes was determined using a two-by-two comparison, and a bar chart was created to illustrate the enrichment level of the pathways with both differential metabolites and differential genes. The horizontal axis of the bar graph represents the metabolic pathway, where red on the vertical axis represents the enrichment *p*-value associated with the differential gene, and green represents the enrichment *p*-value associated with the differential metabolite, which is represented by the −log(*p*-value). The higher the location on the vertical axis, the higher the degree of enrichment (Figure 4). By comparing the DEGs and DEMs in each group, we observed that many were enriched in the same KEGG pathway (Appendix A). The main pathway was the phenylpropanoid pathway.

In the comparison of E2 vs. E9, 5878 differentially expressed genes were found, of which 2527 were upregulated and 3351 were downregulated unigenes. In the comparison of E13 vs. E9, 7686 differentially expressed genes were found, of which 3365 were upregulated and 4321 were downregulated. In addition, we analyzed and characterized these differential genes using KEGG annotation and found that the differential genes were mostly enriched in metabolic pathways and secondary metabolite biosynthesis. Most phenolic acids and flavonoids are synthesized through the phenylpropane biosynthesis pathway, and we focused on the phenylpropane pathway, which was enriched with 120 and 85 DEGs in the two comparisons, respectively. Regarding differential genes in the phenylpropane pathway, we identified structural genes enriched in the phenolic acid and flavonoid biosynthesis pathways, including four MmPALs, two MmC4Hs, three Mm4CLs, four MmCOMTs, three MmCHSs, two MmCHIs, two MmF3Hs, and three MmF3′Hs. We analyzed the gene expression levels of these 23 structural genes using heatmaps (Figure 5). The expression of these structural genes were found to be lower at 900 m altitude than at 200 and 1300 m. Based on the metabolome results, the same expression trend was found for ferulic and caffeic acids, and we hypothesized that these structural genes may regulate the synthesis of ferulic and caffeic acids.

To determine the correlation between the DEMs and DEGs, a gene Pearson’s correlation analysis was performed (Pearson correlation coefficient > 0.8 or <0.8, *p*-value < 0.05). Many metabolites were found to be positively or negatively regulated by multiple genes (Appendix A). Commonly, multiple metabolites are regulated by a single gene or a single metabolite is regulated by multiple genes. In addition, the DEGs significantly correlated with phenolic acids and flavonoids (Figure 6), where the gene-E3N88_30635 (MYB 111) was significantly negatively correlated with 3,4-dihydroxybenzoic acid. These findings indicate the existence of a complex regulatory mechanism between metabolite accumulation and gene expression abundance in *M. micrantha*.

## 3. Discussion

*M. micrantha* is an invasive plant that destroys local cash crops, and its large number of flowers makes its spread difficult to stop [25]. Flower bud differentiation is the most important stage in the development of flowering plants and is an indication of the transition from nutritional to reproductive growth [26]. Therefore, the aim of this study was to control the flowering of *M. micrantha* and thus influence its growth and spread. Metabolites are the end products of cellular bioregulatory processes, the levels of which are a response of plant growth and development to genetic and environmental changes [21]. Bernier proposed that before the initiation of flowering, various growth-regulating substances and metabolites in the body are responsible for promoting or inhibiting the flowering process in plants, which need to act in coordination with each other at the correct concentration and time to promote the transition from nutritional to reproductive growth in the stem tip’s meristem growth [27]. In recent years, the screening of metabolites and related genes using high-throughput methods such as transcriptomics and metabolomics has revealed the mechanisms through which plant secondary metabolites are produced. These methods have also enabled us to link the structure and regulatory genes of key metabolic pathways with the results of metabolomics analyses, to identify key candidate genes, and to construct key gene regulatory networks [28,29,30,31]. Therefore, in this study, we used a combination of metabolomics and transcriptomics to analyze the differences in metabolites during the flower bud differentiation of *M. micrantha* and the differential expression of genes in its metabolic pathways at different altitudes in order to identify the key metabolites controlling flowering in *M. micrantha*.

The primary secondary metabolites in plants are phenolic acids and flavonoids [32]. The results of this study showed that the secondary metabolites in *M. micrantha* flower buds were mainly phenolic acids and flavonoids (Figure 1B). In plants, the accumulation of phenolic acids and flavonoids promotes or inhibits floral developmental processes, such as the process of seed germination. The studies of BI et al. have shown that sinapic acid esters might regulate the ABA-mediated inhibition of dormancy breakage, germination, and growth in *Arabidopsis* [33]; petunia stigmas accumulate kaempferol to increase their seed content [34]. During reproduction, the pollen tube must pass through long distances through the flower tissue to fertilize the ovules [35]. Additionally, flavonols are necessary for the growth of pollen tubes in some crops such as petunias, tobacco, tomato, and corn [36,37,38]. During flower development, in *lisianthus*, flavonol concentration decrease [39]. Gallic acid, protocatechuic acid, (+)-catechin, chlorogenic acid, caffeic acid, and *p*-coumaric acid are the components found in rose petals. These compounds are part of the flavonoid and phenolic acid classes of phytochemicals. During the progression of flower development in roses, the concentrations of these compounds tends to decrease [40,41].

Our results showed that the lowest phenolic acid content occurred in plants at an altitude of 900 m. The expressions of ferulic acid, caffeic acid, and benzoic acid were much lower at 900 m than at 200 and 1300 m (Figure 5A). The reduction in the contents of these substances may promote bud differentiation in *M. micrantha*, so that the number of flowers is higher at 900 m than at 200 and 1300 m. The decrease in the content of these substances may promote bud differentiation in *M. micrantha*.

The differences in the metabolic components in plants can be attributed to variations in gene expression within the phenylpropanoid biosynthesis pathway [42,43,44]. The phenylpropanoid biosynthesis pathway and the synthesized phenolic acids are plant-specific and play a crucial role in plant growth and development [45,46]. In this study, the results of joint transcriptomic and metabolomic analyses revealed that the differential metabolites in *M. micrantha* flower buds were mainly enriched in the metabolic pathway, the biosynthetic pathway of secondary metabolites, and the biosynthetic pathway of phenylpropane (Figure 4), indicating that *M. micrantha* in the bud differentiation stage is metabolically active, and sprouting is accompanied by the production of a large number of secondary metabolites. The flavonoid biosynthesis pathway starts with the production of 4-coumaroyl-CoA, which is derived from the amino acid phenylalanine through a series of enzymatic reactions [47]. The main precursors of plant flavonoid synthesis are 4-coumaroyl-CoA and malonyl-CoA, which undergo a series of enzymatic reactions to produce various flavonoid compounds. One of the key enzymes involved in this process is chalcone synthase (*CHS*) [48,49]. In our study, the phenylpropanoid biosynthetic pathway was found to be significantly different in the combined transcriptome and metabolome analysis, and the differential genes mainly included *PAL*, *4CL*, *F5H*, and *COMT*. Multiple genes in the phenylpropanoid biosynthesis pathway play a role in driving metabolic flow towards phenolic acid synthesis. *PAL*, *4CL*, and *CMOT* were significantly upregulated at E2 and E13 compared with those at E9, and the expression of *PAL* was significantly lower at 900 m than at the other two altitudes. As such, we speculated that *PAL* may positively regulate phenolic acid synthesis in the phenolic acid biosynthesis pathway (Figure 5A).

Transcription factors are involved in the regulation of gene expression by binding to specific DNA sequences known as cis-acting elements, which are located in the promoter regions of genes, activating or repressing transcription through their interactions with each other and with other related proteins and playing a key role in the entire metabolic pathway in plants [50]. MYB-like transcription factors are one of the largest families of plant transcription factors, which are widely involved in plant development and metabolic regulation. Among these, the R2R3-MYB subfamily is mainly involved in the regulation of the phenylpropanoid metabolic pathway [51]. In this study, 43 MYB genes were identified, which will be analyzed in subsequent studies.

## 4. Materials and Methods

### 4.1. Plant Materials

The Tongbiguan Nature Reserve, Dehong Dai and Jingpo Autonomous Prefecture, Yunnan Province, China, has a subtropical monsoon climate with abundant light and heat, abundant rainfall, and a short frost period. *M. micrantha* flower buds were collected on the same day from three 20 cm × 20 cm sample plots at three different altitudes: (1) E2: 200 m altitude; (2) E9: 900 m altitude; and (3) E13: 1300 m altitude. Three replicates were set up for each sample plot. During flower bud collection, a mixture of at least 10 flower buds of *M. micrantha* was used as one biological replicate representing three independent replicates at each altitude. The flower buds were immediately frozen in liquid nitrogen after collection and stored at −80 °C until the total RNA and metabolites were extracted. The *M. micrantha* plants collected at different altitudes were in the inflorescence primordia stage.

### 4.2. Metabolite Detection and Analysis

The metabolome analysis was performed by Wuhan Maiwei Biotechnology Co., Ltd. (Wuhan, China) (https://www.metware.cn/Home11.html accessed on 6 March 2021) using a widely targeted metabolome method. The flower buds of *M. micrantha* from the three different altitude groups were dehydrated using a vacuum freeze-dryer and crushed into a powder to preserve the metabolites present in the samples. A portion of the lyophilized powder (100 mg) was weighed and extracted with 1.2 mL of 70% methanol. The extraction was performed overnight at 4 °C to ensure the efficient extraction of the metabolites from the sample. After extraction, the samples were centrifuged at 12,000 rpm for 10 min to separate the solid particles from the liquid supernatant, which contained the extracted metabolites. The supernatant was filtered (0.22 µm pore size; ANPEL, Shanghai, China). A scheduled multiple-reaction monitoring method was used to quantify the metabolites [52]. The filtrate was analyzed using a UPLC-ESI-MS/MS system [53]. Linear ion trap and triple quadrupole scans were acquired with a UPLC-MS/MS system, which was connected to an ESI Turbo Ion-Spray interface controlled using Analyst 1.6.3 software (AB Sciex, Framingham, MA, USA) and was operated in both the positive and negative ionization modes. The filtered metabolite data were analyzed with Analyst 1.6.3 software for an unsupervised principal component analysis (PCA). A multivariate principal component analysis (PCA) was conducted to identify the patterns and trends in the metabolome data. This analysis helped us to visualize the similarities and differences between the samples based on their metabolite profiles. Metabolites with a VIP score ≥ 1 and fold-change ≥ 2 or ≤0.5 were considered differentially accumulated metabolites (DEMs). The identified DEMs were subjected to functional annotation and enrichment analysis based on the KEGG database (https://www.kegg.jp/kegg/compound/ accessed on 6 May 2022). This analysis helped us to understand the biological pathways and processes associated with the differentially accumulated metabolites.

### 4.3. Transcriptome Sequencing and Analysis

As with the transcriptome study, all samples used for the metabolome analysis were in the same stage and from the same location. A PureLink Plant RNA Kit (Invitrogen, Carlsbad, CA, USA) was used to extract the total RNA. An Agilent 2100 bioanalyzer was used to determine the quality of the RNA. mRNA was isolated from the total RNA through the use of magnetic beads with poly-T oligo attached. The cDNA library was constructed, and transcriptome sequencing was performed on an Illumina sequencing platform (Illumina HiSeqTM 2500)( (Illumina Inc., San Diego, USA)). The raw data obtained from sequencing were filtered to remove sequences containing only sequenced splice sequences. Sequences with an N content exceeding 10% of the number of bases in the read and sequences with low-quality (Q ≤ 20) bases exceeding 50% of the number of bases in the read were excluded. The raw sequences were quality-controlled to obtain clean sequences by removing splice sequences, low-quality sequences, multiple N sequences, and sequences that were too short. Then, the sequencing base error rate and GC base content were calculated. The expression levels of the genes were calculated using fragments per kilobase per million mapped fragments (FPKM) [54], and DESeq2 [55,56] was used to analyze the differentially expressed genes between two samples in different periods (fold change ≥ 1 and q < 0.05) [57]. The annotation information of differentially expressed genes was obtained via comparison with the Kyoto Encyclopedia of Genes and Genomes (KEGG, https://www.genome.jp/kegg accessed on 6 June 2022) database, a well-known and reliable database for interpreting the molecular-level details of genomes, enzymes, and chemicals in organisms [58].

### 4.4. Conjoint Analysis of the Transcriptome and Metabolome Data

From the quantitative analysis, specific genes and metabolites were identified as differentially expressed, meaning their expression levels significantly differed between the compared conditions or groups. These DEGs and DEMs were the focus of further analysis, as they may be important for understanding the underlying biological processes. The metabolites and their associated transcripts were then mapped to the relevant metabolic pathways in the KEGG database. The cor function in the R statistical programming language was used to calculate the Pearson correlation coefficients between the differentially expressed genes (DEGs) and the differentially expressed metabolites (DEMs). Finally, the network diagrams representing the correlations between the DEGs and DEMs were plotted using Cytoscape 3.8.2 software. The network diagrams helped us to understand the interconnectedness and potential regulatory relationships between the DEGs and DEMs.

## 5. Conclusions

In this study, based on a combined transcriptomic and metabolomic approach, we investigated the differences in the metabolites and genes of *M. micrantha* during flower bud differentiation at different altitudes, and we identified a total of 658 differential metabolites, including 161 flavonoids and 78 phenolic acids. Candidate genes affecting the synthesis of flavonoids and phenolic acids in *M. micrantha* buds, including 23 structural genes in the phenylpropane synthesis pathway and a transcription factor gene-E3N88_30635 (*MYB 111*), were also explored, providing a new avenue for the study of the molecular mechanism of the accumulation of phenolic acids and flavonoids in *M. micrantha* buds. This study also has reference value for the subsequent use of bioengineering methods to regulate the flowering of *M. micrantha*.

## Figures and Tables

**Figure 1 ijms-24-16550-f001:**
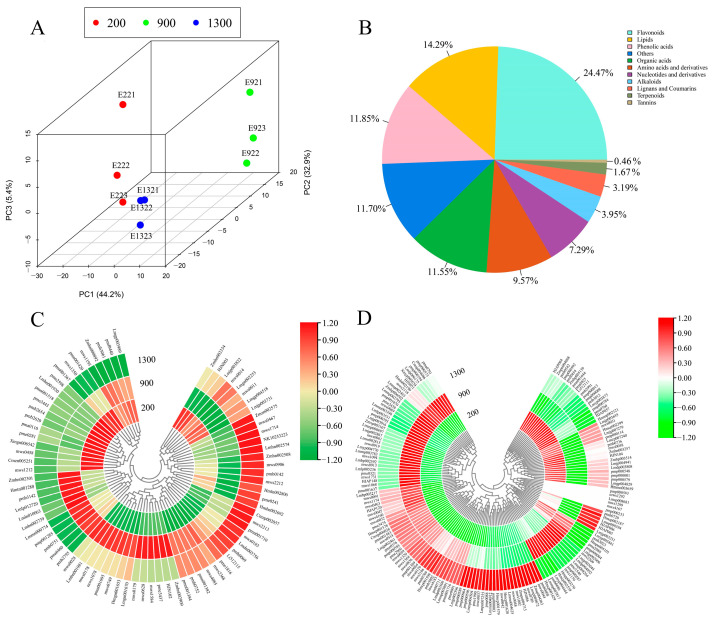
Differential metabolites analysis: (**A**) PCA 3D plot; (**B**) classification of the 658 metabolites; (**C**) phenolic acids and (**D**) flavonoids contents at three different altitudes. Red in the heatmap denotes a considerable buildup of phenolic acid and flavonoids; green denotes a significant decrease in their levels.

**Figure 2 ijms-24-16550-f002:**
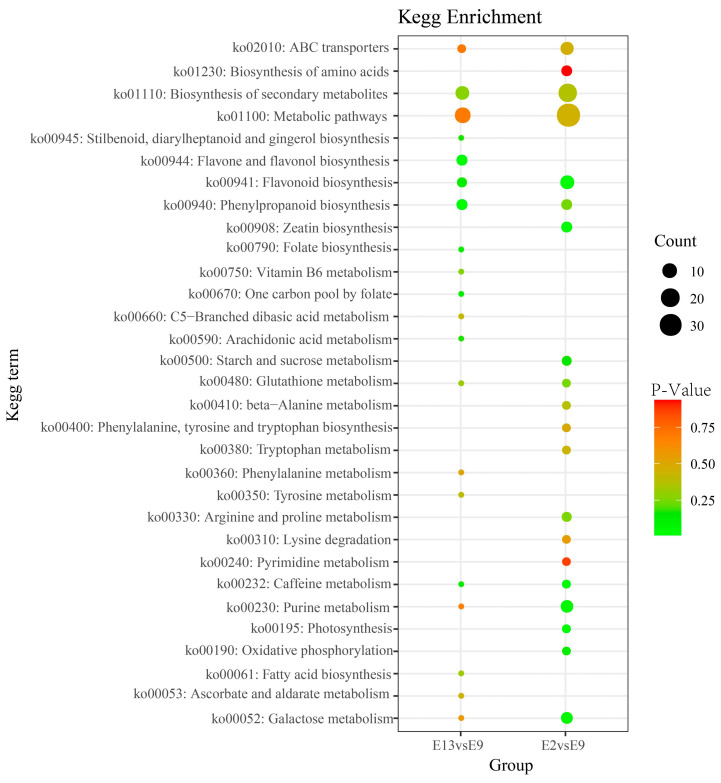
KEGG enrichment analysis of DEMs between the comparison groups (E13 vs. E9, E2 vs. E9). Each bubble represents a metabolic pathway. A larger bubble indicates a stronger impact factor. Bubble colors represent the *p*-values of the enrichment analysis.

**Figure 3 ijms-24-16550-f003:**
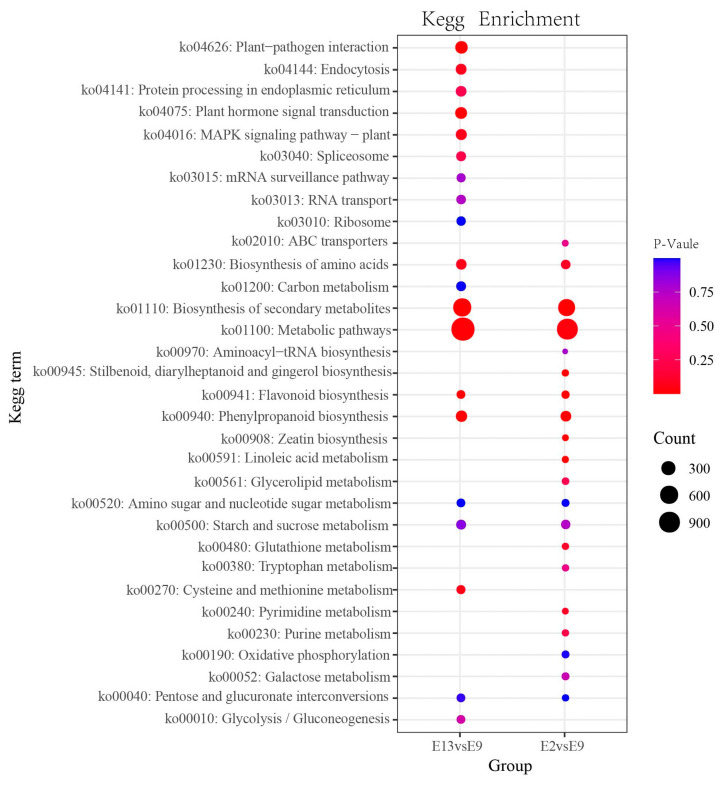
KEGG enrichment analysis of DEGs between the comparison groups (E13 vs. E9, E2 vs. E9). Each bubble represents a metabolic pathway. A larger bubble indicates a factor with a stronger impact. Bubble colors represent the *p*-values of the enrichment analysis.

**Figure 4 ijms-24-16550-f004:**
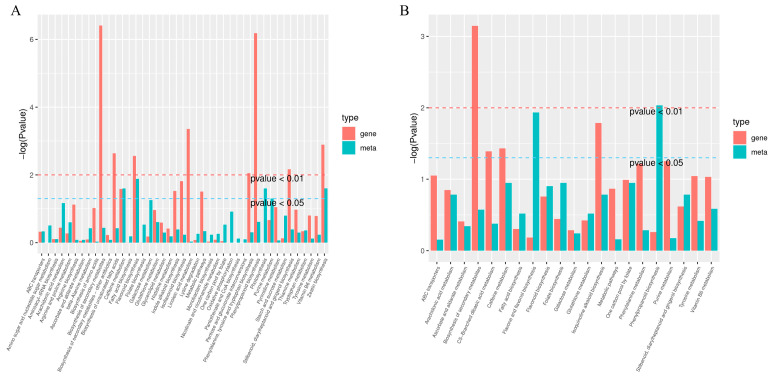
Differentially accumulated metabolites (green column) and differentially expressed genes (red column) that were enriched in the same pathway. (**A**) E2 vs. E9 KEGG enrichment analysis histograms. (**B**) E13 vs. E9 KEGG enrichment analysis histograms.

**Figure 5 ijms-24-16550-f005:**
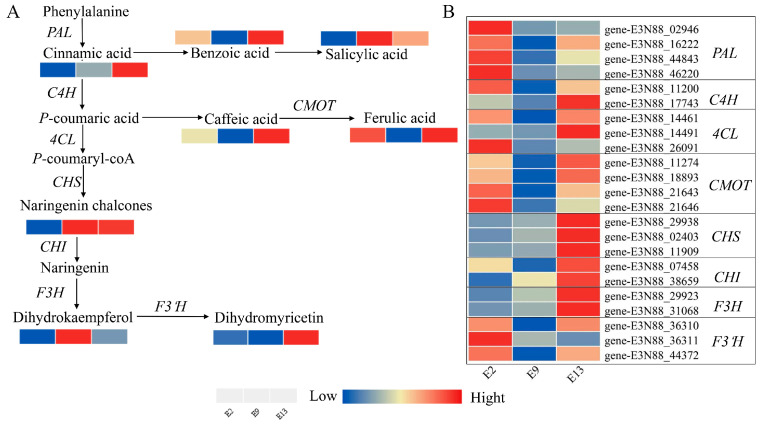
Reconstruction of the phenylpropanoid biosynthetic pathway. (**A**) Phenylpropane metabolic pathway map; the relative content of the metabolite is indicated by the colored cells below it, where the redder the cells are, the higher the quantity is. (**B**) Expression of related genes in the phenylpropane pathway according to FPKM values. PAL: phenylalanine ammonia-lyase, C4H: cinnamate 4-hydroxylase, 4CL: 4-coumarate-CoA ligase, CMOT: caffeoyl-CoA O-methyltransferase, CHS: chalcone synthase, CHI: chalcone isomerase, F3H: flavanone-3-hydroxylase, F3′H: flavonoid 3′-hydroxylase.

**Figure 6 ijms-24-16550-f006:**
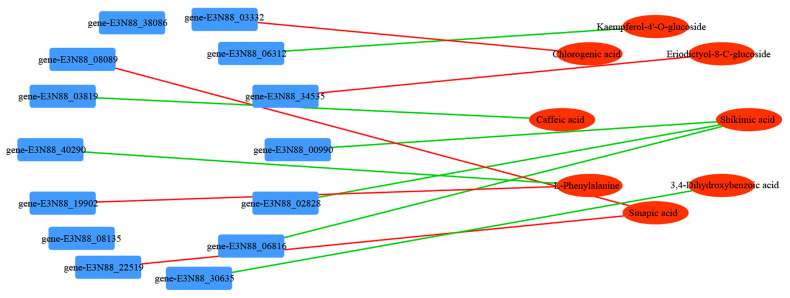
Connection network between differentially expressed genes (blue) and differentially accumulated metabolites (red). Red lines represent positive correlations and green lines represent negative correlations.

## Data Availability

All data generated or analyzed during this study are included in this published article and its Appendix A. The nucleotide sequences of the raw data from this study were submitted to the NCBI Sequence Read Archive (SRA) under the accession number PRJNA792910.

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
