# Peer review of "Phenolic Acids and Flavonoids Play Important Roles in Flower Bud Differentiation in Mikania micrantha: Transcriptomics and Metabolomics"

_ijms, 2023, doi:10.3390/ijms242316550_

Round 1
Reviewer 1 Report
Comments and Suggestions for Authors
This manuscript by Ling Pei and colleagues covers an interesting topic related with potential control of the Mikania micrantha which is an invasive vine that could induce the kill of several vegetal species. Authors show a great number of potential valuable results for a further control study related with this specie. However, in my opinion, the work could be improve introducing a few minor changes in several points before a possible publication.
1 – I recommend making a final English revision. There are a few mistakes.
2 – Line 113: Which metabolic “… significant involvement in the metabolic changes observed” means ? This is not totally clear.
3 – Line 224-240: Authors are not discussing the results. Rewrite or delete this sentence.
4 – Line 244: Which “… important roles ….” ?
5 – Line 252: (+)-catechin; p-coumaric acid “p” is in italic form.
6 – Line 293-299: Rewrite and all this paragraph must be introduced in conclusions.
7 – Could you introduce more details related with these climatic conditions ?
8 – PCA analysis was carried out using which software ?
Comments on the Quality of English LanguagePlease see comments for quality of English language:
I recommend making a final English revision. There are a few minor mistakes.
Reviewer 2 Report
Comments and Suggestions for Authors
In this study a combination of metabolomics and transcriptomics methods was used to analyze the accumulation patterns in the flower buds of M. micrantha. This is a perennial herbaceous or woody vine of in the Asteraceae family, and is one of the damaging weeds. In a previous article (reference 23) it was discovered that the number of florets and inflorescences at 900 meters altitude was considerably greater than on 200- and 1300-meter altitude. This study now investigated the gene expression and metabolic differences of M. micrantha at altitudes (”treatment”) on 200 m, 900 m, and 1300 m to gain insight into the genetic and metabolites in inflorescences and florets.
The Introduction of the paper give an overview for the motivation of the study. The part M&M must be improved, because are some missing information, especially of the samplings. Using this kind of methods (metabolomics and transcriptomics), the presentation of the Results is shown in a known manner. Compared to 900 m, 200 m and 1300 m exhibited greater levels of gene expression. The expression levels of DEGs must be clarified (see line 161). The study revealed that the regulation of flavonoids and phenolic acids influenced the accumulation patterns of these metabolites. It was "observed", that the canopy suppressed the accumulation on 900 m of most flavonoids and phenolic acids. The focus of considerations was the phenylpropanoid biosynthetic pathway. Missing are information about the environmental factors (plant community, temperature, soil, etc.; see file) and samplings.
The results are discussed based on the available and suitable literature. However, there is no critical analysis of the results and differences that were determined at the altitude of 900 m. "An effort to discover new ways to stop the spread of M. micrantha" (line 82) is not shown. In the Conclusions "These results are important for revealing the molecular mechanism of flower bud differentiation (not observed!) in M. micrantha, and provide new ideas for M. micrantha control and flowering regulation studies" is very broad and also general.
Comments/remarks, see file.

Moderate editing of English language required.
Round 2
Reviewer 2 Report
Comments and Suggestions for Authors
The strong revision of this manuscript was successful! This is now an interesting paper with new insight at the stage of inflorescence primordial differentiation.